# Outcomes of Returning Medically Actionable Genomic Results in Pediatric Research

**DOI:** 10.3390/jpm12111910

**Published:** 2022-11-16

**Authors:** Amy A. Blumling, Cynthia A. Prows, Margaret H. Harr, Wendy K. Chung, Ellen Wright Clayton, Ingrid A. Holm, Georgia L. Wiesner, John J. Connolly, John B. Harley, Hakon Hakonarson, Michelle L. McGowan, Erin M. Miller, Melanie F. Myers

**Affiliations:** 1Cincinnati Children’s Hospital Medical Center, Cincinnati, OH 45229, USA; 2Children’s Hospital of Philadelphia, Philadelphia, PA 19104, USA; 3Department of Pediatrics, Columbia University, New York, NY 10032, USA; 4Division of Genetic Medicine, Department of Medicine, Vanderbilt University Medical Center, Nashville, TN 37232, USA; 5Division of Genetics and Genomics, Boston Children’s Hospital, Boston, MA 02115, USA; 6Department of Pediatrics, Harvard Medical School, Boston, MA 02115, USA; 7US Department of Veterans Affairs Medical Center, Cincinnati, OH 45220, USA; 8Department of Pediatrics, Perelman School of Medicine, University of Pennsylvania, Philadelphia, PA 19104, USA; 9Biomedical Ethics Research Program, Department of Quantitative Health Sciences, Mayo Clinic, Rochester, MN 55905, USA; 10Department of Women’s, Gender, and Sexuality Studies, College of Arts and Sciences, University of Cincinnati, Cincinnati, OH 45221, USA; 11Department of Pediatrics, College of Medicine, University of Cincinnati, Cincinnati, OH 45267, USA

**Keywords:** genomic screening, pediatric genomics, return of genomic results, healthcare outcomes

## Abstract

Purpose: The electronic Medical Records and Genomics (eMERGE) Phase III study was undertaken to assess clinical utility of returning medically actionable genomic screening results. We assessed pediatric clinical outcomes following return of pathogenic/likely pathogenic (P/LP) variants in autosomal dominant conditions with available effective interventions. Methods: The two eMERGE III pediatric sites collected outcome data and assessed changes in medical management at 6 and 12 months. Results: We returned P/LP results to 29 participants with outcome data. For 23 of the 29 participants, the P/LP results were previously unknown. Five of the 23 participants were already followed for conditions related to the P/LP variant. Of those receiving novel results and not being followed for the condition related to the P/LP result (*n* = 18), 14 (77.8%) had a change in healthcare after return of results (RoR). Following RoR, cascade testing of family members occurred for 10 of 23 (43.5%). Conclusions: The most common outcomes post-RoR included imaging/laboratory testing and health behavior recommendations. A change in healthcare was documented in 77.8% of those receiving results by 6 months. Our findings demonstrate how return of genomic screening results impacts healthcare in pediatric populations.

## 1. Introduction

Identifying disease risk and implementing available surveillance and disease prevention strategies during childhood provides an opportunity to study the effectiveness of such interventions on future health. Several studies have explored the benefits of genomic sequencing in pediatric patients based on family risk or clinical indication, but few have explored what happens following return of genomic screening results in larger pediatric populations without a clinical indication [1,2,3,4,5].

Genomic screening in the adult population is increasingly useful in identifying adults without clinical symptoms who are at an increased risk of preventable health conditions, including early onset cancers and heart disease [6,7,8]. When adults receive pathogenic or likely pathogenic (P/LP) variant screening results, they can make better-informed decisions with their providers regarding healthcare management, such as increased frequency of monitoring for disease, preventive lifestyle, medications, or surgeries, and decisions regarding reproduction [6,7,8].

Current pediatric recommendations focus on offering opportunistic or secondary genomic testing when clinical genomic sequencing is ordered for a child presenting with a specific clinical indication or a family history of a disease that is medically actionable in childhood (i.e., can benefit from intervention in childhood and/or adolescence) [9,10]. Proposed benefits of opportunistic testing in a pediatric clinical setting include the potential to identify at-risk family members [9,10,11]. However, there is limited information about the clinical utility of population-based genomic screening and subsequent healthcare outcomes in the pediatric population [4,5]. While data are emerging on the impact of return of genomic screening results in the pediatric population [4,5,12,13], a practice and knowledge gap remains in how providers and patients receiving medically actionable genomic screening results in pediatric populations implement healthcare changes following the return of such results.

To address this gap, the electronic Medical Records and Genomics (eMERGE) Phase III study included the return of genetic testing results to pediatric participants. The eMERGE Network is a consortium of multiple sites funded by the National Institutes of Health that develops and disseminates research utilizing biorepositories, electronic medical records, and genomic medicine. While Phase I, beginning in 2007, focused on demonstrating the utility of combining genomic data with longitudinal electronic health record (EHR) data [14], each subsequent phase increased emphasis on genomic medicine implementation. During Phase III, each site designed and implemented studies that used a Network developed sequencing panel for testing [15] and returned results to participants [16]. While the Network consisted of 10 sites, only two chose to enroll children, Cincinnati Children’s Hospital Medical Center (CCHMC) and Children’s Hospital of Philadelphia (CHOP). The set of returnable actionable genomic results varied across the 10 sites in accordance with local institutional policies [17,18]. This report is focused on changes in healthcare among pediatric eMERGE Phase III participants following return of P/LP variants in genes associated with autosomal dominant conditions.

## 2. Materials and Methods

Each of the ten eMERGE Phase III clinical sites designed and implemented site-specific RoR projects [16,19]. Network-wide outcomes measures, including laboratory/imaging studies ordered, referrals, other healthcare changes post-RoR, cascade testing of family members, were collected centrally using disease-specific outcomes forms [19]. The focus of this report is secondary analysis of the centrally collected outcome data completed by the two pediatric sites, CCHMC and CHOP.

## 3. Pediatric Participants

There were three different cohorts across the two pediatric sites. CCHMC had two cohorts: a prospective adolescent cohort and a biobank cohort. The adolescent cohort was a prospectively enrolled cohort of adolescents aged 13–17 years at CCHMC. Adolescents were not selected based on a clinical indication for genetic testing. The adolescent-parent dyad first made independent choices followed by joint choices about the type of genetic results they wanted to learn about the adolescent [20,21], and results were returned that matched their joint choices [12].

Both CCHMC and CHOP also had biobank cohorts. The biobank cohorts had participants whose parents had previously given permission for their child to participate in their respective biobanks. During the biobank consent process at each institution, parents indicated if they were willing to be recontacted about clinically actionable results identified during a study that used their child’s DNA for research purposes [22]. At CCHMC, only biobank participants who would be <18 years old at the time of RoR were included. At CHOP, all participants were enrolled in the biobank when they were <18 years old. If they had turned 18, they were reconsented as adults prior to RoR.

## 4. Results Eligible for Return

All eMERGE III participants’ DNA samples were analyzed with a multi-gene panel of 109 genes and over 1500 single nucleotide variants specifically created for Network discovery projects and RoR projects [15]. Each clinical site identified a subset of genes associated with clinically actionable conditions for their IRB-approved RoR studies.

The subset of genes selected for potential return differed between the two pediatric sites as well as between CCHMC’s two cohorts, as previously detailed in Hoell et al., 2020 [18]. In brief, for CCHMC’s adolescent cohort, CCHMC prospectively enrolled adolescent-parent dyads to make choices to learn about all, some, or none of 32 possible conditions informed by 84 genes [20,21]. Choices included learning results for a limited number of adult-onset conditions (hereditary breast, ovarian and colon cancers) and carrier status for autosomal recessive disorders.

For the CCHMC biobank cohort, analysis of actionable genes was limited to those that informed risk for conditions that can manifest during childhood and for which disease prevention and/or early detection and treatment are available during childhood. If a P/LP result was identified in a child’s biobanked sample, a letter was mailed to the child’s parent notifying the parent that their child’s stored DNA was used in a genetic research study and giving them options to learn their child’s result. Non-responders to the initial letter were contacted up to two additional times via additional letters.

For the CHOP biobank cohort, participants were contacted for potential result disclosure only when a P/LP variant was identified in a gene that informed risk for conditions that can manifest during childhood and for which disease prevention and/or early detection and treatment are available during childhood, such as hypercholesterolemia, malignant hyperthermia, and others. Eligible participants were contacted up to 3 times by telephone and once by letter before considered lost to follow-up. Biobank participants at CHOP who had a P/LP variant in a gene for an adult-onset condition were contacted directly only if they re-consented into the biobank as an adult and had agreed to be recontacted for potential result disclosure.

CCHMC and CHOP implemented similar result return procedures for biobank participants. Genetic counselors returned P/LP variant results by telephone to parents of biobank participants < 18 years. At CHOP, RoR occurred directly with pediatric biobank participants who had turned 18 years old and reconsented. Unique to the CCHMC adolescent prospective cohort, telephone disclosure involved both the parent and adolescent participants. Genetic counselors recommended that participants come to the hospital for relevant screening or further genetic counseling after RoR.

## 5. Outcome Data

Investigators across the Network created disease-specific outcome forms for each gene-disease association [19]. The outcomes forms were compared to outcomes developed by the ClinGen Actionability working group and showed concordance in variables to assess as outcomes in eMERGE III [4]. Specifically, eMERGE III outcomes forms focused on process outcomes such as referrals, a specialist visit, laboratory test or ECG consequent to learning the P/LP result, and cascade testing for potentially affected family members. Research assistants and genetic counselors at pediatric sites manually reviewed their participants’ electronic health records (EHRs) to collect and complete eMERGE outcome data forms at 6 and 12 months after return of a P/LP variant in a gene that informed disease risk.

## 6. Secondary Data Analysis

A secondary codebook was developed by the primary author that included existing variables recorded in the outcome forms, as well as new variables created specifically for the current RoR analysis. Existing variables from the original eMERGE III outcomes forms included: demographics, gene variant and associated condition, family cascade testing after returning results, whether the variant identified in eMERGE III was known to the parent or the child prior to participation in the genetic research study, laboratory/imaging studies, medications ordered, and referrals. Outcome variables created specific to our secondary analysis directly related to the P/LP variant included presence of variant in EHR post-RoR and additional recommended health behaviors directly related to the P/LP variant. Additionally, if no action was taken (e.g., “no” was entered in the outcome form indicating no evidence of laboratory/imaging studies, medications, and/or referrals being ordered), we assessed the appropriateness of inaction based on the test result received and current practice guidelines.

An assessment of overall healthcare change for the participant (i.e., whether certain outcome variables demonstrated a change in care occurred) was evaluated at both 6 and 12 months. If any laboratory/imaging studies, medications, referrals, or additional recommended health behaviors were present and directly related to RoR, overall healthcare change was coded as “Yes.” All outcomes were directly associated with returned results.

Statistical frequencies and counts are reported below for both pediatric institutions returning results. Additionally, we developed case exemplars for four participants to highlight specific outcomes related to RoR for participants.

## 7. Results

In total, from both CCHMC and CHOP, 29 P/LP variant results were returned to individuals who also had outcome forms as described below and demonstrated in Figure 1:

**CCHMC prospective adolescent cohort:** CCHMC prospectively enrolled 161 adolescents between the ages of 13–17 and a parent/legal guardian into the adolescent cohort. One adolescent’s DNA sample from this cohort could not be sequenced. Results from the remaining 160 were reported. Six P/LP results informing disease risk were available for RoR. Of these six, one participant refused RoR, and two results were for a gene (*CHEK2*) that did not have an associated network outcomes form. The three remaining P/LP results returned to adolescent/parent dyads in this cohort had available outcome forms.

**CCHMC biobank cohort:** CCHMC shipped 2840 deidentified biobank samples for discovery studies. Of these samples, 91 deidentified P/LP results were received, 45 of these were from adult participants and excluded from this pediatric study. The remaining forty-six available results were for children under the age of 18 and thus, their parents were eligible to be invited to participate in our study. Three parents declined recontact during the biobank informed consent process, one participant was deceased and of the remaining 42 eligible parents who were mailed letters, three were returned to sender and one child had turned 18 after the first letter was mailed and did not reconsent as an adult to the biobank. Nineteen parents of children desired RoR, but one could not be scheduled. The remaining 18 P/LP results were returned to parents for whom outcome forms were available.

**CHOP biobank cohort:** CHOP shipped 3020 deidentified biobank samples for discovery aims and 1984 of these were pediatric samples potentially eligible for RoR. P/LP deidentified result reports were received for 101 participants and 52 of these were for children < 18 years at biobank consent. Sixteen were excluded from Outcomes study because P/LP result informed risk for an adult onset condition; 2 were deceased and 14 declined consent for RoR study upon biobank recontact. Of the remaining 20 eligible for RoR, 8 had results returned within time frame for outcomes study. CHOP consented 8 participants with children between the ages of 12–26 (at time of RoR) from an existing pediatric biobank repository for return of newly associated results. Three adults (>18 years at time of RoR) were enrolled in the biobank as children (<18 years), reconsented as adults, and results returned directly to them. All 8 had P/LP variants returned and an associated outcome form available.

Of the 29 participants from both sites, P/LP variants associated with the following conditions were returned: increased risk for arrhythmia, aortopathy, breast cancer, cardiomyopathy/heart failure, malignant hyperthermia, hypercholesterolemia, tuberous sclerosis complex, chronic kidney disease, Fabry disease (female), and CACNA1A associated neuropathologies (Table 1). The most common category of P/LP variants returned was cardiomyopathy (*n* = 9 P/LP; Table 1). All participants were enrolled into the CCHMC prospective adolescent cohort or pediatric biobank repositories prior to the age of 18. Participant ages at study RoR ranged from 4–26 years, with a mean age of 14.1 years. Three participants were 18 years or older at RoR, while the remainder were under 18 years. There were twice as many males who received P/LP results compared to females (*n* = 20 M vs. *n* = 9 F) (Table 2). The majority of participants with results returned identified as White (*n* = 17, 58.6%) (Table 2).

Of the 29 for whom outcome forms were available, five participants already knew of the variant returned through the study. Variants that were previously known included the following: TSC1, RYR1, KCNQ1, HNF1B, GLA One additional participant with a P/LP DSP variant was a former NICU patient receiving care for a variety of congenital conditions, and it was not possible to ascertain whether medical management changes were related to the previously known genetic variant or the newly ascertained variant as part of the eMERGE III study. These six participants were excluded from the outcome analysis.

Of the remaining 23 participants with no previous knowledge of their variant, five were already being followed clinically for a condition that could be associated with the P/LP variant (Table 3). For example, one of the participants who received a P/LP LDLR variant was currently being treated for hypercholesterolemia, prior to the variant being identified. Demographic data for the group of 23 are displayed in Table 2.

**Six-month outcomes data.** Six-month outcome data, including whether tests were ordered, medications prescribed, the P/LP variant was acknowledged in the participant’s medical records, and others, are detailed in Table 3.

Of the 18 participants who were not previously followed for a clinical phenotype related to the P/LP variant, more than half had some type of laboratory test or imaging study performed after RoR (*n* = 11, 61.1%). Additionally, more than half received post-RoR referrals to condition-specific specialists (*n* = 11, 61.1%). For 12 of 18 participants (66.7%), providers recommended health behaviors or provided education for the participants, including: use or avoidance of specific hormone-mediating oral contraceptives, diet/exercise changes, education on signs and symptoms of possible disease progression, annual exams or imaging studies, medical alert bracelets, and anesthesia precautions. New medications were ordered for only one participant (Table 3).

Seven of the 18 participants had no laboratory or imaging studies, or medications ordered after RoR. For two of the seven (28.6%), laboratory or imaging studies, or medications were not clinically indicated based on the gene returned. For example, breast cancer monitoring is not clinically indicated for an adolescent < 18 years receiving a P/LP *BRCA1* result.

When examining the EHRs of the participants, documentation of the P/LP variant or mention of the condition was found in the primary problem list post-RoR for 12 of 23 participants (Table 3). An additional five (21.7%) had the variant acknowledged or listed elsewhere in their EHR (e.g., provider notes or encounters, lab tests, etc.), while the remaining six (26.1%) did not have the variant/condition listed anywhere in their EHR, aside from a letter from a genetic counselor at the time of RoR.

At 6 months post-RoR, a healthcare change had been initiated for 14 of the 18 (77.8%) who had not previously known their P/LP variant and were not being treated for a clinically related phenotype. A healthcare change included any laboratory/imaging studies ordered, medications prescribed, referrals made, or additional health behaviors recommended.

**Twelve-month outcome data.** Twelve-month post-RoR outcome data included the same variables as 6-month data for participants with new P/LP variant results. At twelve months post-RoR, only six of 18 participants had data in the EHR and/or outcomes forms for testing, referrals, or medications ordered. Of the six participants with 12-month data, two participants had new laboratory testing or imaging studies performed related to the genetic finding (33%); one of these two participants also had a new medication ordered by 12 months. The same two participants also received new referrals. An additional two participants (33%) did not have any healthcare changes noted in the EHR. Finally, the two remaining participants were previously being followed for a clinical condition that could have been associated with their condition prior to RoR, so we were unable to determine if any changes at 12 months were due to RoR or the previously treated condition.

Overall, at 12 months, two of 18 had a healthcare change for themselves or family members that differed from the 6-month outcomes. However, the majority of healthcare changes had already occurred by the 6-month time point.

**Cascade Testing.** Of the 23 participants, cascade testing was documented for all adolescent participants (*n* = 3) and 35% (*n* = 7) of biobank participants. One of the participants who had cascade testing was previously being followed for a clinical phenotype related to their P/LP variant. Eight biobank participants had no data related to cascade testing in the outcome forms or EHR. Reasons for undergoing or not undergoing cascade testing were not assessed. This study was not able to capture cascade testing completed outside the study window of 12 months post-RoR or testing of family members that was completed at outside institutions and not documented in our participants’ EHR.

## 8. Case Exemplars

**Case exemplar 1.** A 7-year-old African American female biobank participant had a pathogenic variant in *KCNQ1* identified, consistent with increased risk for long QT syndrome. No previous signs or symptoms of arrhythmias had been documented in the patient. No associated family history was listed in EHR. Identification of this variant led to cascade testing in mother, father, and brother, with father and brother also sharing the genetic variant. The participant completed post-RoR EKG and stress test, demonstrating continuous sinus rhythm and borderline QT prolongation during exercise. Referrals were made to a cardiac electrophysiologist and a cardiac genetic counselor. Medication was not initiated based on EKG findings. Additional health behaviors recommended included education on QT prolongation and recommendations for follow up EKG and ECHO.

**Case exemplar 2.** A 16-year-old African American male biobank participant had a pathogenic variant in *TNNI3*, consistent with an increased risk for cardiomyopathy. The participant had been followed for obesity and systemic hypertension prior to identification of the *TNNI3* variant. No associated family history was listed in the EHR. Identification of the variant led to cascade testing in both the mother and father, with the mother also heterozygous for the variant. Prior to RoR, the participant had an EKG and echocardiogram due to a diagnosis of hypertension. Post-RoR, the participant had another EKG and echocardiogram. Neither indicated a need for medication. Referrals were made to a cardiologist and cardiac genetic counselor. The participant was not prescribed any therapies and did not receive any clinical diagnoses. Additional health behaviors recommended included education on cardiomyopathy, recommendations for diet and exercise, and possible hypertensive management in the future.

**Case exemplar 3.** A 16-year-old White male adolescent cohort participant had a pathogenic variant in *SCN5A* identified, consistent with an increased risk for Brugada syndrome and associated arrhythmias. He had an uncertain history of presyncope and syncope. Identification of this variant led to cascade testing in his mother and brother. His mother was also found to be heterozygous for the genetic variant. The participant’s mother had a history of seizure of unknown etiology. There was also a maternal family history of sudden death. The participant completed post-RoR EKG and loop/event monitor, demonstrating sinus bradycardia with first degree AV block and nonspecific intraventricular conduction delay. Findings were consistent with a diagnosis of Brugada syndrome. His mother was also subsequently diagnosed with Brugada syndrome. During the 6–12-month outcomes window, the participant received an implantable cardiac monitor and then experienced syncope that correlated with a long sinus pause. This event prompted the placement of an implantable cardioverter defibrillator.

**Case exemplar 4.** A 17-year-old White male biobank participant had a pathogenic variant in *SCN5A*, consistent with increased risk for arrhythmia. The participant had a past medical history of craniosynostosis and normal EKG prior to surgery, but there was no knowledge of a P/LP *SCN5A* variant. There were no data in the EHR or outcome form about cascade testing. The participant completed post-RoR EKG with no abnormal findings. Referral was made to a cardiologist. The mother of the participant was adopted and did not know her family history.

## 9. Discussion

To our knowledge, this is one of the first studies to assess medical outcomes based on electronic health records documentation after return of genomic screening results in pediatric settings. Most healthcare changes took place within 6 months of receiving results. A change in healthcare for the participant was documented by 6 months in 77.8% of those receiving P/LP results. The most common 6-month outcome post-RoR was a recommendation for change in one or more health behaviors (66.7%). This was followed closely by laboratory/imaging studies and subspeciality referrals (61.1%).

A challenge with exploring the health outcomes using EHR data following RoR is that we could assess provider recommendations and orders, but actual patient behaviors are less well-documented, especially those such as dietary, exercise, or lifestyle changes. Additionally, if action should have been taken based on current practice guidelines, but was not, we cannot determine if it was a missed opportunity, the participant had not returned to CCHMC or CHOP during the follow-up period, or the participant had received care elsewhere, but it was not documented in the pediatric hospital’s EHR.

The detailed case exemplars in the current study suggest that for some individuals, the RoR of a P/LP variant had a significant impact on overall health and healthcare. One individual was able to find an end to their diagnostic odyssey related to occasionally losing consciousness, as this was found to be a symptom of Brugada syndrome caused by a variant in SCN5A. Although there were significant changes documented in some of the participants, for many participants, it is unclear what effect the RoR had on their overall health. One of the concerns regarding RoR in screening or secondary findings in pediatric research is the potential for regret after learning results. While we did not measure this in the current study, our previous work in Lillie, et al., has demonstrated that even with RoR of P/LP variants resulting in major health or healthcare changes, no participants expressed regret after receiving their result [12].

One of the proposed reasons for returning clinical exome and genome sequencing secondary findings or predictive testing results is to identify health issues in other family members, especially the parents of children with P/LP genomic findings [9,10]. This is suggested as being one of the few ways to identify parents with the same variant, which is in the best interest of both the child and parent [9,10]. Less than half of both the CCHMC and CHOP participants had evidence of cascade genetic testing to identify other family members with the same P/LP variant at 12 months post-RoR. Other studies have also reported limited uptake of cascade testing by family members. The Clinical Sequencing Exploratory Research (CSER) Consortium study exploring secondary findings in clinical genomic sequencing interviewed a subset (*n* = 18) of their 74 participants who received P/LP secondary findings to explore sharing of results with family members and subsequent cascade testing. Of their 18 participants, all reported that they had disclosed their variant to at least one family member; however, only four participants were able to confirm that their family members had undergone cascade testing within 12 months post RoR. The remaining individuals reported that other family members did not undergo cascade testing, or they were unsure whether they had [13]. If cascade testing or other healthcare changes are desired outcomes secondary to return of genomic screening in the pediatric population, future studies should collect data on reasons for pursuing or not pursuing cascade testing, as well as risks, benefits, and outcomes of cascade testing. With the expansion of telemedicine, there are new opportunities to support families with education and cascade genetic testing centrally through research studies and/or services that support nationwide genetic testing.

Previous studies have reported wide variability in healthcare change recommendations and subsequent participant follow-through on changes post-RoR to adults [6,13]. Of the 6240 participants in the CSER study with genomic sequencing, secondary findings were disclosed to 74 participants [13]. All participants reported that they felt an increased interest in pursuing recommended healthcare follow-up actions post-RoR; however, it is unknown if participants actually engaged in these recommended actions [13]. In the 2008 Geisinger MyCode Community Health Initiative, chart reviews of 23 individuals receiving a P/LP variant linked to increased risk for familial hypercholesterolemia after genomic screening revealed that 78% were prescribed lipid-lowering therapy post-RoR, but only 22% met their LDL-C goal reduction [6]. Longitudinal studies focused on pediatric participants is needed to understand long-term health outcomes after learning secondary findings or genomic screening results. In Phase IV, the eMERGE Network is requiring all funded sites to enroll both children and adults, but assessment of health outcomes remains limited to 6 and 12 months [23].

## 10. Limitations

One of the limitations of this study is that the outcome assessment only documented actions taken by providers at the research site. Other limitations include some differences in the types of results returned between the two pediatric institutions and small sample sizes, which limit generalizability. Finally, it is unclear if returning these results improved participants’ health, as data were only collected during a timeframe of 6 to 12 months. Changes outside of this time frame or with other healthcare institutions may have occurred, but they were not part of our outcome assessment.

## 11. Conclusions

Our study has demonstrated that genomic screening in the pediatric eMERGE sites resulted in an increase in healthcare actions taken by providers for participants with newly diagnosed P/LP variants. Overall, 77.8% of unaffected participants receiving novel P/LP findings and not previously being followed for a phenotype related to the P/LP finding had a change in healthcare provided. The most frequent actions were ordering laboratory or imaging studies, referrals to specialists, and recommendations for health behavior changes. Cascade testing was relatively low in this study, with less than half of study participants documented as having family members tested. If cascade testing is one of the primary goals of return of genomic results in pediatric findings, then researchers and clinicians may need to emphasize this when disclosing results. With the limited information available on clinical utility or impact of returning genomic screening results in pediatric patients and families, this study helps elucidate how such RoR impacts changes in healthcare management in this population.

## Figures and Tables

**Figure 1 jpm-12-01910-f001:**
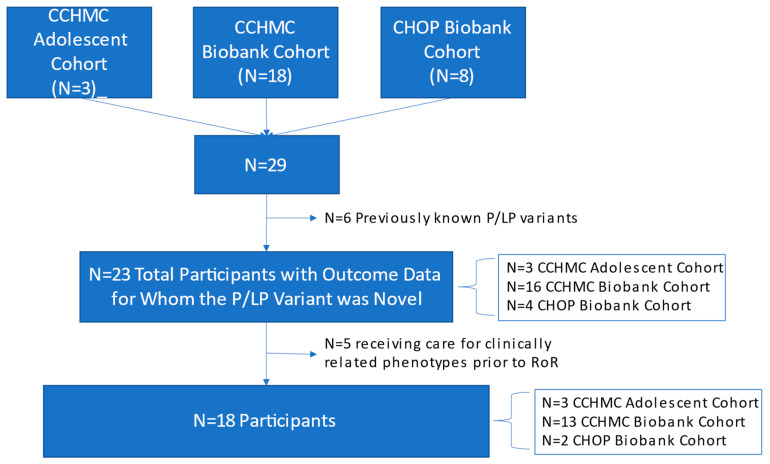
Participants with P/LP Results and Outcome Data by Site and Cohort.

**Table 1 jpm-12-01910-t001:** Variants and Number Returned with Outcomes Forms Available.

Condition Category	Specific P/LP Gene Variant	*n*
Aortopathy	*SMAD3*	1
Arrhythmia	*KCNQ1*	2
	*SCN5A*	2
Breast/Ovarian Cancer—Women	*BRCA1*	1
Cardiomyopathy	*DSP*	1
	*MYBPC3*	3
	*MYH7*	3
	*PKP2*	1
	*TNNI3*	1
Chronic Kidney Disease/Maturity Onset Diabetes of the Young (MODY)	*HNF1B*	1
Fabry Disease	*GLA*	1
Malignant Hyperthermia	*RYR1*	2
Hypercholesterolemia	*APOB*	3
	*LDLR*	4
Tuberous Sclerosis Complex	*TSC1*	1
CACNA1A associated neuropathologies	*CACNA1A*	1
Multiple endocrine neoplasia	*RET*	1
Total		*n* = 29

**Table 2 jpm-12-01910-t002:** Participant Demographics.

Variable	*n* (%) Total Participants with Outcomes Forms	Total Participants with Outcomes Forms for Whom the P/LP Was Novel	Total Participants with Outcomes Forms for Whom the P/LP Was Novel and Were Not Being Followed for a Clinically Related Phenotype
Cohort	*n* = 29	*n* = 23 *	*n* = 18 **
CCHMC Adolescent	3	3	3
CCHMC Biobank	18	16	13
CHOP Biobank	8	4	2
Age range (years)	4–26 (mean 14.1)	4–20 (mean 13.7)	4–20 (mean 14)
Sex			
Male	20 (69%)	17 (73.9%)	13 (72.2%)
Female	9 (31%)	6 (26.1%)	5 (27.8%)
Race			
White	17 (58.6%)	14 (60.9%)	11 (61.1%)
Black/AA	9 (31%)	7 (30.4%)	5 (27.8%)
Multiple races	2 (6.9%)	1 (4.3%)	1 (5.6%)
Asian	1 (3.4%)	1 (4.3%)	1 (5.6%)

* Excludes 6 who already knew of variant; ** Excludes 5 who were already being followed for disease.

**Table 3 jpm-12-01910-t003:** Six-month outcomes.

Variable	Yes	No
Labs/Imaging/Tests *n* = 18 *	11 (61.1%)	7 (38.9%)
Medications ordered *n* = 17 ***	1 (5.5%)	16 (88.9%)
Referrals ordered *n* = 18 *	11 (61.1%)	7 (38.9%)
Add’l Health Behavior ** *n* = 18 *	12 (66.7%)	6 (33.3%)
Total with Healthcare Change *n* = 18	Yes *n* = 14 (77.8%)	No *n* = 4 (22.2%)

* Excludes 5 participants already being followed for phenotype or condition that could be associated with variant; ** Add’l Health Behavior: Includes provider recommended health behaviors of use or non-use of specific hormone-mediating oral contraceptives, diet/exercise changes, education on signs and symptoms of possible disease progression, annual exams or imaging studies, medical alert bracelets, and anesthesia precautions; *** Data was missing in outcome forms and/or EHR on medications ordered for 1 participant.

## Data Availability

Data is not available due to privacy.

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
