# Peer review of "Outcomes of Returning Medically Actionable Genomic Results in Pediatric Research"

_jpm, 2022, doi:10.3390/jpm12111910_

Round 1
Reviewer 1 Report
This manuscript reports results regarding clinical utility following the disclosure of actionable genomic variants to children and families who were tested largely without a clinical indication. The research is timely and important. The manuscript is well-written.
I have only several minor suggestions for the authors. On line 177, the authors report that 42 parents of minors in the CCHMC cohort were contacted regarding return of results but only 18 responded. This seems to be a significant observation. It merits an additional sentence or two about what efforts were made to contact the families. The Methods section only mentions that a letter was sent to families with an option to learn the results. Were any additional actions taken? If not, it might be noted in the discussion that, in a clinical context, a more vigorous effort would be made to contact families about actionable results. A bit more discussion of this relatively low uptake in the discussion section seems warranted.
Line 236: The use of "such actions" is a little confusing as it was initially unclear whether it refers to the lack of action by the participants.
Lines 362-364: The authors note that "More research is needed..." The EMERGE consortium was an enormous undertaking and, although important, the project yielded only a small sample size for this particular dataset. It would be helpful for the NIH, other sponsors, and investigators for the authors to expand this recommendation to suggest what types of research will be necessary to further define the clinical utility of this sort of genomic testing.
Author Response
Reviewer 1
- On line 177, the authors report that 42 parents of minors in the CCHMC cohort were contacted regarding return of results but only 18 responded. This seems to be a significant observation. It merits an additional sentence or two about what efforts were made to contact the families. The Methods section only mentions that a letter was sent to families with an option to learn the results. Were any additional actions taken? If not, it might be noted in the discussion that, in a clinical context, a more vigorous effort would be made to contact families about actionable results. A bit more discussion of this relatively low uptake in the discussion section seems warranted.
- Thank you for this comment. We have clarified two additional attempts were made to reach non-responders. This edit can be found on lines 137-138 and 143-144.
- Line 236: The use of "such actions" is a little confusing as it was initially unclear whether it refers to the lack of action by the participants.
- Thank you for this comment. We have clarified “such actions” as meaning laboratory studies, imaging studies, or medications. This edit can be found on lines 274-276.
- Lines 362-364: The authors note that "More research is needed..." The EMERGE consortium was an enormous undertaking and, although important, the project yielded only a small sample size for this particular dataset. It would be helpful for the NIH, other sponsors, and investigators for the authors to expand this recommendation to suggest what types of research will be necessary to further define the clinical utility of this sort of genomic testing.
- Thank you for this comment. We have clarified the type of additional research needed, such as targeted research exploring health outcomes and exploring longitudinal research. We have also added some ways the current eMERGE Network is meeting some of these gaps. These edits can be found in lines 408-414.
Reviewer 2 Report
Overall this is an important study and the data is presented in an excellent paper. I have no real criticisms of the paper or study, but i was left with a number of questions that may or may be able to be answered/addressed.
- The overall numbers are small, which leads to one question - what % of pediatric age patients in these 2 cohorts had LP/P secondary findings?
- Why were the 3 adolescents tested? Out of how many tested
- Same for the bio bank samples - how many were tested?
- How was the consent done for the bio bank samples?
- Why only 2/10 sites included pediatric samples? All sites have robust pediatric centers.
- do we think these %’s would hold up if for example universal genomic sequencing were implemented on every pediatric patient (which I. Assume is the loft long term vision)? And if we did that, would this process remain the same? I doubt it, so how would this process proceed? Perhaps these latter points are outside the scope, but the subject prompts the mind to go there.
- while many geneticists are familiar with EGMERGE, and there are cited papers on it, perhaps a few more sentences describing it (phases 1, 2, and 3), would be good.
- I was surprised by the low number that had cascade testing. We’re these patients who from the bio banks (which I assume were likely collected longer ago, and maybe with less stringent discussions). Any ideas why? Also, was cascade testing free through the study? Did (lack of) insurance play a role in those who did not follow through?
Some if not all of my questions may be outside this study, but it does prompt one to think. Perhaps acknowledging these unknowns is worth a sentence or 2.
Author Response
Reviewer 2
- The overall numbers are small, which leads to one question - what % of pediatric age patients in these 2 cohorts had LP/P secondary findings?
- Thank you for this comment. We have clarified the number of initially enrolled participants and percentages of P/LP in the CCHMC prospective adolescent cohort. These edits can be found on lines 188-197. Additional information for the biobank cohorts are addressed in question 3.a below.
- Why were the 3 adolescents tested? Out of how many tested
- Thank you for this comment. With the response to comment 1.a, we believe we have addressed this question.
- Same for the bio bank samples - how many were tested?
- Thank you for this comment. We have clarified the number of samples tested, P/LP results received, and final participants who received their results. These edits can be found in lines 200-211 for CCHMC and in lines 212-218 for CHOP.
- How was the consent done for the bio bank samples?
- Thank you for this comment. We believe we have answered this question in lines 112-119.
- Why only 2/10 sites included pediatric samples? All sites have robust pediatric centers.
- Thank you for this question. At the time of the Phase III eMERGE study, it was not required for sites to return pediatric results. These edits can be found in lines 83-85.
- do we think these %’s would hold up if for example universal genomic sequencing were implemented on every pediatric patient (which I. Assume is the loft long term vision)? And if we did that, would this process remain the same? I doubt it, so how would this process proceed? Perhaps these latter points are outside the scope, but the subject prompts the mind to go there.
- Thank you for this comment. We agree this is an interesting question, although it was outside the scope of our study. However, we hope others will pursue this line of questioning in the future.
- while many geneticists are familiar with EGMERGE, and there are cited papers on it, perhaps a few more sentences describing it (phases 1, 2, and 3), would be good.
- Thank you for this comment. We have added descriptions of eMERGE and associated phases. These edits can be found in lines 76-89.
- I was surprised by the low number that had cascade testing. We’re these patients who from the bio banks (which I assume were likely collected longer ago, and maybe with less stringent discussions). Any ideas why? Also, was cascade testing free through the study? Did (lack of) insurance play a role in those who did not follow through?
- Thank you for this comment. We have clarified the number of participants who had or did not have cascade testing. These edits can be found on lines 302-313.